# Lymphatic filariasis epidemiology in Samoa in 2018: Geographic clustering and higher antigen prevalence in older age groups

**Colleen L. Lau**[1]*, **Kelley Meder**[1], **Helen J. Mayfield**[1], **Therese Kearns**[2], **Brady McPherson**[1], **Take Naseri**[3], **Robert Thomsen**[3], **Shannon M. Hedtke**[4], **Sarah Sheridan**[5], **Katherine Gass**[6], **Patricia M. Graves**[7]

**1** Research School of Population Health, Australian National University, Canberra, Australia, **2** Menzies School of Health Research, Charles Darwin University, Brisbane, Australia, **3** Ministry of Health, Apia, Samoa, **4** Department of Physiology, Anatomy and Microbiology, La Trobe University, Bundoora, Victoria, Australia, **5** School of Public Health and Community Medicine, University of New South Wales, Sydney, Australia, **6** Neglected Tropical Diseases Support Center, The Task Force for Global Heath, Decatur, Georgia, United States of America, **7** College of Public Health, Medical and Veterinary Sciences, James Cook University, Cairns, Queensland, Australia

* colleen.lau@anu.edu.au

**Data Availability Statement:** All relevant data are within the paper. We are unable to provide individual-level antigen prevalence data and

## Abstract

### Background

Samoa conducted eight nationwide rounds of mass drug administration (MDA) for lymphatic filariasis (LF) between 1999 and 2011, and two targeted rounds in 2015 and 2017 in North West Upolu (NWU), one of three evaluation units (EUs). Transmission Assessment Surveys (TAS) were conducted in 2013 (failed in NWU) and 2017 (all three EUs failed). In 2018, Samoa was the first in the world to distribute nationwide triple-drug MDA using ivermectin, diethylcarbamazine, and albendazole. *Surveillance and Monitoring to Eliminate LF and Scabies from Samoa (SaMELFS Samoa)* is an operational research program designed to evaluate the effectiveness of triple-drug MDA on LF transmission and scabies prevalence in Samoa, and to compare the usefulness of different indicators of LF transmission. This paper reports results from the 2018 baseline survey and aims to i) investigate antigen (Ag) prevalence and spatial epidemiology, including geographic clustering; ii) compare Ag prevalence between two different age groups (5–9 years versus ≥10 years) as indicators of areas of ongoing transmission; and iii) assess the prevalence of limb lymphedema in those aged ≥15 years.

### Methods

A community-based cluster survey was conducted in 30 randomly selected and five purposively selected clusters (primary sampling units, PSUs), each comprising one or two villages. Participants were recruited through household surveys (age ≥5 years) and convenience surveys (age 5–9 years). Alere Filariasis Test Strips (FTS) were used to detect Ag, and prevalence was adjusted for survey design and standardized for age and gender. Adjusted Ag prevalence was estimated for each age group (5–9, ≥10, and all ages ≥5 years) for random

demographic data because of the potential for breaching participant confidentiality. The communities in Samoa are very small, and individual-level data such as age, sex, and village of residence could potentially be used to identify specific persons. For researchers who meet the criteria for access to confidential data, the data are available on request from the Human Ethics Officer at the Australian National University Human Research Ethics Committee, email: human.ethics. officer@anu.edu.au.

**Funding:** This work received financial support from the Coalition for Operational Research on Neglected Tropical Diseases (OPP1053230, CL, https://www.ntdsupport.org/cor-ntd), which is funded at The Task Force for Global Health primarily by the Bill & Melinda Gates Foundation, by the United States Agency for International Development through its Neglected Tropical Diseases Program, and with United Kingdom Aid from the British people (OPP1053230, CL). KG was an employee of The Task Force for Global Health. CLL was supported by an Australian National Health and Medical Research Council (www. nhmrc.gov.au) Fellowship (Grant number 1109035). Other than KG who was an employee of The Task Force for Global Health and has been included as an author, the funders had no role in study design, data collection and analysis, decision to publish, or preparation of the manuscript.

**Competing interests:** The authors have declared that no competing interests exist.

and purposive PSUs, and by region. Intraclass correlation (ICC) was used to quantify clustering at regions, PSUs, and households.

## Results

A total of 3940 persons were included (1942 children aged 5–9 years, 1998 persons aged ≥10 years). Adjusted Ag prevalence in all ages ≥5 years in randomly and purposively selected PSUs were 4.0% (95% CI 2.8–5.6%) and 10.0% (95% CI 7.4–13.4%), respectively. In random PSUs, Ag prevalence was lower in those aged 5–9 years (1.3%, 95% CI 0.8–2.1%) than ≥10 years (4.7%, 95% CI 3.1–7.0%), and poorly correlated at the PSU level (R-square = 0.1459). Adjusted Ag prevalence in PSUs ranged from 0% to 10.3% (95% CI 5.9–17.6%) in randomly selected and 3.8% (95% CI 1.3–10.8%) to 20.0% (95% CI 15.3–25.8%) in purposively selected PSUs. ICC for Ag-positive individuals was higher at households (0.46) compared to PSUs (0.18) and regions (0.01).

## Conclusions

Our study confirmed ongoing transmission of LF in Samoa, in accordance with the 2017 TAS results. Ag prevalence varied significantly between PSUs, and there was poor correlation between prevalence in 5–9 year-olds and older ages, who had threefold higher prevalence. Sampling older age groups would provide more accurate estimates of overall prevalence, and be more sensitive for identifying residual hotspots. Higher prevalence in purposively selected PSUs shows local knowledge can help identify at least some hotspots.

### Author summary

Lymphatic filariasis (LF), a disease caused by infection with worms transmitted by mosquitoes, has long been present in Samoa. Since the 1960s, Samoa has attempted to control the disease through many rounds of annual administration of two deworming drugs to the whole population. However, Samoa recently observed that LF transmission was still occurring, prompting mass drug administration (MDA) with three drugs in 2018. Here, we report the baseline survey of an operational research program to evaluate the triple-drug MDA. The survey assessed prevalence and geographical distribution of LF in the population, compared prevalence by age groups, and investigated the burden of elephantiasis (swollen limbs caused by long-term LF infection). The study confirmed ongoing transmission, with 4% of those aged ≥5 years showing antigen in their blood as evidence of infection. Antigen prevalence was more than three times higher in those aged ≥10 years (4.7%) than in 5–9 year-old children (1.3%). Infection was highly clustered within households and villages, with up to 20% of residents infected in known hotspot villages. Future surveillance strategies should consider that i) testing older age groups would provide more accurate indication of LF transmission, and ii) local knowledge can help identify transmission hotspots.

## Introduction

Lymphatic filariasis (LF) is a neglected tropical disease (NTD) caused by three species of filarial worms (*Wuchereria bancrofti*, *Brugia malayi*, and *B. timori)*, and transmitted between humans by a range of mosquito species including *Aedes*, *Culex*, and *Anopheles* [1]. Once a mosquito passes LF larvae into a new host's bloodstream, the worms reach the lymphatic system where they mature into adults, mate, and release microfilariae (Mf) [2]. Adult worms can live for 5–7 years and release millions of Mf [2]. While most infected individuals remain asymptomatic, some will develop swollen limbs, and scrotal hydrocele in males [3]. Not only do individuals with these chronic complications endure disfigurement and disability, they often also experience social stigmatization and suffer from mental health issues and negative economic effects [4]. LF causes significant disease burden globally, and contributes to more than 5.8 million Disability Adjusted Life Years (DALYs) [5].

Current global estimates suggest that LF affects over 120 million people in 72 countries, mostly in Africa, Asia, the Western Pacific, and limited areas in the Americas [1,6]. In 1997, the World Health Organization (WHO) established the Global Programme to Eliminate Lymphatic Filariasis (GPELF) [7,8], which aims to eliminate LF as a public health problem by implementing two key strategies: i) interrupting transmission via large-scale treatment of populations in endemic areas through mass drug administration (MDA), and ii) providing care to alleviate suffering for those with chronic complications. Initially, recommended drugs for MDA included the use of a 2-drug regimen of either ivermectin and albendazole (in areas where LF is co-endemic with onchocerciasis), or diethylcarbamazine and albendazole (in other areas) [9].

To determine whether there is evidence of ongoing LF transmission, WHO currently recommends Transmission Assessment Surveys (TAS), which use critical threshold numbers of antigen (Ag) positive children aged 6–7 years in population-representative surveys (generally school-based) in defined evaluation units (EUs). Critical cut-off values are calculated so that the likelihood of an evaluation unit passing is at least 75% if true Ag prevalence is 0.5%, and no more than 5% if the true Ag prevalence is ≥1% [10]. At least two TAS are recommended; the first to be conducted >6 months after the last round of MDA to determine if MDA can be stopped, and additional TAS to confirm the absence of transmission [10]. TAS is a widely used and accepted tool for post-MDA surveillance, but recent studies have identified that TAS might not be sensitive enough for identifying localized areas of ongoing transmission (hotspots), particularly if there is significant spatial heterogeneity in infection prevalence, e.g. as shown in previous studies in Samoa [11] and American Samoa [12–14].

Since 2000, over 7.7 billion MDA treatments have been delivered to >910 million people living in 68 LF-endemic countries [1]. By 2018, 14 countries had been validated by the WHO as having eliminated LF as a public health problem, but 893 million people in 49 LF-endemic countries still needed MDA [1]. These included countries that had not commenced MDA, not completed MDA in all endemic areas, or not achieved elimination targets despite completing the recommended rounds of MDA. Although the two-drug regimen had been successful in many places, it became apparent that augmented treatment regimes and/or other strategies would be needed to successfully eliminate LF globally. In November 2017, WHO endorsed the use of two annual rounds of triple-drug MDA (ivermectin, diethylcarbamazine, albendazole: IDA), which has been shown to be more effective for achieving sustained clearance of microfilariae compared to two-drug regimes [9]. The new triple-drug MDA is recommended for countries where onchocerciasis is not endemic and in areas that i) have not yet started MDA, or ii) have provided fewer than four effective rounds of MDA, or iii) where MDA results have been suboptimal [9].

Samoa is a tropical island country in the South Pacific where LF is endemic. *Wuchereria bancrofti* is the only species of filarial worm that has been reported to cause LF in Samoa, and infection is transmitted by multiple species of *Aedes* mosquitoes, predominantly *Aedes polynesiensis*. Ten rounds of MDA were distributed between 1964 and 1998 prior to GPELF [15], and a further six rounds between 1999 and 2007 under the Pacific Programme to Eliminate Lymphatic Filariasis (PacELF), the regional program which aimed to support the achievement of GPELF goals in the 22 Pacific Islands Countries and Territories [16,17]. While LF Ag prevalence (by the rapid Alere ICT) in Samoa was greatly reduced from 4.4% in 1998 to 1.1% in 2004, the WHO-recommended threshold for stopping MDA was not reached [18,19] and further nationwide MDAs were distributed in 2008 and 2011 [20,21].

Samoa is divided into four administrative regions: Apia Urban area (AUA), Northwest Upolu (NWU), Rest of Upolu (ROU), and Savai'i (SAV). For TAS, three EUs were defined: AUA/ROU, NWU, and SAV (two regions were combined to achieve the most pragmatic grouping by population size and expected prevalence). In 2013, TAS in school children aged 6–7 years identified that transmission was still occurring in NWU, and two more rounds of two-drug MDA were distributed in NWU in 2015 and 2017 (Samoa Ministry of Health). A repeat TAS in 2017 showed that all three EUs failed to meet elimination targets and further nationwide MDA was required [22]. In 2017, Samoa prepared a National Action Plan to Eliminate LF, and in August 2018, Samoa was the first country in the world to implement nationwide triple-drug MDA using IDA [22,23].

The *Surveillance and Monitoring to Eliminate Lymphatic Filariasis and Scabies from Samoa (SaMELFS Samoa)* project was designed as an operational research program to monitor and evaluate the effectiveness of two rounds of nationwide triple-drug MDA on LF transmission in Samoa. This paper focuses on the baseline human survey conducted in 2018, seven years after the last nationwide MDA and one year after the last MDA in NWU. In this paper, we aim to i) investigate Ag prevalence and spatial epidemiology, including any areas of high transmission (hotspots), ii) compare the Ag prevalence between two different age groups (5–9 years versus ≥10 years) as indicators of areas of ongoing transmission, and iii) assess the prevalence of limb lymphedema in those aged ≥15 years.

## Methods

### Ethics statement

Ethics approvals were granted by the Samoan Ministry of Health and The Australian National University Human Research Ethics Committee (protocol 2018/341). The study was conducted in close collaboration with the Samoa Ministry of Health, the WHO country office in Samoa, and the Samoa Red Cross.

This study was designed and implemented using culturally appropriate methods, and consent was sought at multiple levels. Firstly, village leaders were contacted to seek permission for their community's involvement and asked to inform community members about the study prior to the field team's visits. Secondly, on arrival at each selected household, the field team leader provided a verbal explanation of the study and written information to an adult resident, and sought verbal approval to enter the household. Thirdly, each individual (or a parent/guardian of child participants) provided written informed consent (and verbal assent from minors) prior to enrolment in the study. Fieldwork activities were conducted with bilingual local field teams, and in a culturally sensitive manner. Written information, consent forms, and surveys were provided/conducted in Samoan or English according to each participant's preference.

## Study location

Samoa consists of two main islands, Upolu and Savai'i, and a number of small adjacent islands in the South Pacific. Samoa has a tropical climate, and the majority of the population live in small coastal villages. The larger islands are made up of mountains, valleys, tropical forests, wetlands, fringing reefs and lagoons. In 2018, Samoa had an estimated total population of 201,316 persons, with approximately 19% in AUA, 35% in NWU, 23% in ROU, and 22% in SAV [24].

## Study design

*SaMELFS Samoa* comprises multiple linked studies that included LF epidemiology prior to triple-drug MDA (baseline survey); MDA participation, coverage and adverse events; comparison of the usefulness of different indicators of transmission in different age groups (Ag, Mf, anti-filarial antibodies); using filarial DNA in mosquitoes as indicators of transmission (molecular xenomonitoring); impact of each round of triple-drug MDA on human and mosquito indicators; prevalence of lymphedema; and the impact of ivermectin on scabies prevalence. For the human survey, recruitment for different components of the study was dependent on age: LF Ag, Mf, Ab (≥5 years), lymphedema examination (≥15 years), MDA participation, coverage and adverse events (≥2 years) [23], and scabies prevalence (all ages). This paper focuses on the epidemiology of Ag, Mf, and lymphedema in the 2018 baseline human survey. Results of other study components will be reported in separate publications, and results of Ab prevalence are still pending. (See S1 Checklist for STROBE checklist for cross-sectional studies).

## Primary sampling units

The 2018 baseline human survey was a community-based cross-sectional cluster survey in 35 primary sampling units (PSUs) on Upolu, Savai'i and Manono islands (Fig 1). Thirty PSUs were selected by systematic random sampling, starting from a random point on a line list of 338 villages in the 2016 national census. In eight of the initially selected PSUs, the total population was less than 600, so an additional village (next on the line list) was added to those PSUs to ensure that target sample size in 5–9 year-olds was achievable (see below). In addition, five PSUs were purposively selected by the Ministry of Health as 'suspected hotspots' based on local knowledge and results of previous LF surveys [17]. All purposively selected PSUs consisted of one village. The 35 PSUs therefore included a total of 43 individual villages.

## Target sample size

The target sample size for the Ag prevalence study was 2000 people aged ≥10 years and 2000 children aged 5–9 years, i.e. approximately 57 individuals in each target age group in each of the 35 PSUs. Sample size calculations were based on numbers required to detect a critical threshold of 2% Ag prevalence in each age group, with a 5% chance of type 1 error, 75% power (when true prevalence is 1%), a design effect of 2.0, and correcting for the finite population [10]. The number of households required to recruit 57 participants aged ≥10 years was calculated by assuming an average of 5.18 household members aged ≥10 years [24], and an overall 25% non-response rate due to absence or refusal; this approach determined that 15 households would need to be surveyed in each PSU to achieve target sample size. A much larger number of households would have been required to recruit the required number of 5–9 year-olds. For logistic reasons, 5–9 year-olds were recruited from a combination of household surveys (in the 15 selected households in each PSU), as well as convenience surveys (see below). All household

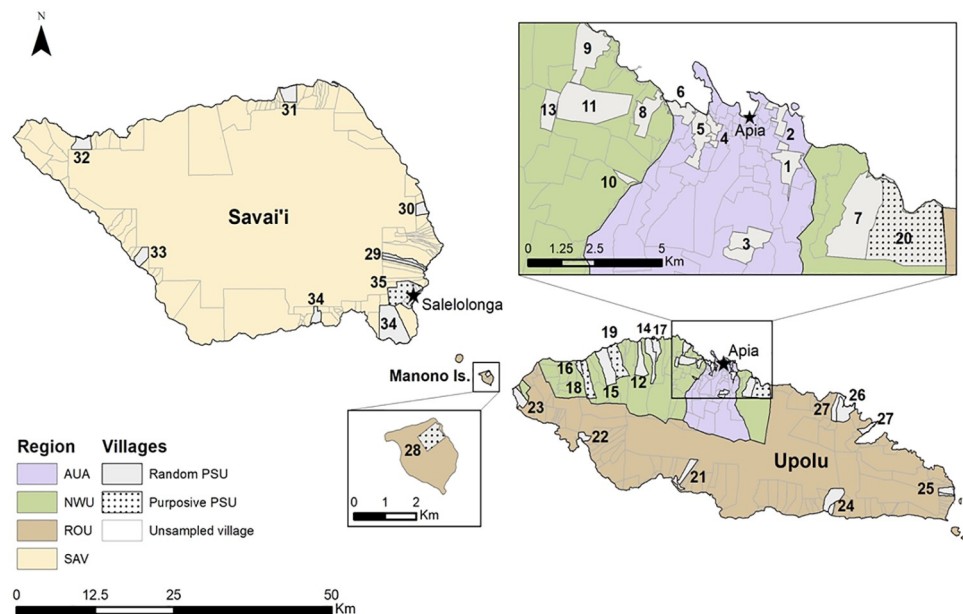

| Region | Apia Urban Area (AUA) | Northwest Upolu NWU) | Rest of Upolu (ROU) | Savai'i (SAV) |
|---|---|---|---|---|
| Number of random PSUs (number of villages) | 6 (9) | 11 (11) | 7 (10) | 6 (8) |
| Number of purposive PSUs (number of villages) | 0 (0) | 3 (3) | 1 (1) | 1 (1) |
| Total PSUs (total villages) | 6 (9) | 14 (14) | 8 (11) | 7 (9) |
| Map reference for random PSUs | 1. Vaivase Tai<br>2. Vaiala Tai + Vaiala Uta<br>3. Avele + Letava<br>4. Fugalei + Vaimea<br>5. Vaimoso<br>6. Vaitoloa | 7. Letogo<br>8. Vaiusu<br>9. Puipaa<br>10. Ululoloa<br>11. Vaitele Fou<br>12. Lotosoa<br>13. Nuu<br>14. Tuanai<br>15. Fasitoo Uta<br>16. Vailuutai<br>17. Leauvaa | 21. Fusi<br>22. Faleseela<br>23. Manono Uta<br>24. Salani + Utulaelae<br>25. Mutiatele + Saleaaumua<br>26. Falefa<br>27. Musumusu + Faleapuna | 29. Lalomalava + Safua<br>30. Lano<br>31. Safotu<br>32. Sataua<br>33. Sagone<br>34. Papa + Tafua |
| Map reference for purposive PSUs | | 18. Fasitoo Tai<br>19. Faleasiu<br>20. Laulii | 28. Salua (Manono Island) | 35. Salelolonga |

**Fig 1. SaMELFS Samoa 2018 survey: Map showing administrative regions and locations of 35 PSUs (43 villages) on Upolu, Savai'i, and Manono Islands, and summary of PSUs and villages by region and selection method (random or purposive).**

members aged ≥15 years were examined for lymphedema of the limbs; no specific sample size calculation was done for lymphedema.

## Household selection

Within each PSU, 15 households were selected prior to the village visits. If a PSU consisted of two villages, the number of households selected in each was proportional to the village populations. Using aerial satellite images from Google Earth, all buildings in each PSU that were

judged to be possibly a house based on roof size (ranging from plantation huts to urban houses) were numbered in order to create a 'virtual walk' that passed every house in the community. From this list of numbered houses, a house was randomly chosen as the starting point, and every nth house was selected where n was the total number of houses in the community divided by 15. The field teams located selected households using a combination of detailed printed village maps, Google Maps, and a smartphone application (Maps.me version 9.5.2/ Google data version 191124) programmed with GPS locations of selected households.

If a selected building was uninhabited or not a domestic residence, it was replaced with the closest inhabited household, or the household of the family who owned the building. If there were multiple houses in close proximity, the team faced north and selected the first house clockwise from there as the closest household. If nobody was home during the first visit, the household was revisited later that day or on another day. If the second visit was unsuccessful, the household was replaced with the closest inhabited household. After the field team had visited or attempted to visit the 15 selected households (or their replacements), if the number of participants sampled was <80% of target sample size, up to five additional households were randomly selected and surveyed. Locations of replaced and additional households were recorded on printed field maps.

## Household survey

Household surveys were generally conducted between 3pm-8pm when people of all ages were most likely to be home. For cultural reasons, surveys were not conducted on Sundays. Individuals were considered a household member if the house was their primary place of residence ('mostly sleep in this house') or if they slept there the previous night. If eligible household members were absent during the survey but expected to return, the field team arranged to revisit the house later in the day or another day. Participants were interviewed by field assistants using standard electronic questionnaires on a smartphone application. One adult in each house was designated as 'household head' and answered general questions such as the number of household members of each age group. Electronic data were collected and managed using the Secure Data Kit (SDK) cloud-based data management system (www.securedatakit.com). GPS coordinates of each household were recorded using the SDK smartphone application.

## Convenience survey

For the convenience survey, 5–9 year-old children were invited by a community leader to attend a central place in the village with a parent or guardian, e.g. at a school, community hall, church, or large fale (Samoan open house). Similar to the household survey, demographic data were collected through a standard electronic questionnaire by interviewing a parent or guardian, but GPS location was not recorded because surveys were not conducted at their place of residence. The target sample size was 57 per PSU, so the number of 5–9 year-old children recruited in the convenience survey depended on the number who had already been tested through the household survey. A second convenience survey was arranged if target sample size was not met. Where the convenience survey occurred prior to the household survey, children already tested were not re-tested if their households were selected for the survey.

## Blood sample collection and processing

From each participant, a finger prick blood sample (up to 400 μL) was collected into a heparin microtainer. The samples were kept cool until they were processed in a field laboratory on the same or next day. The Alere Filariasis Test Strip (FTS) (Scarborough, ME, USA) was used to detect circulating filarial Ag [25]. Using a micropipette, 75μl of heparinized blood was placed

onto the FTS, and assessed after 10 minutes per manufacturer's instructions. If sufficient blood was available, positive tests were repeated to confirm the result. For all blood samples with a positive FTS result, thick blood smears were prepared according to WHO guidelines [10]; three 20μL lines of blood were placed on to a single slide using a pipettor, and up to three slides were prepared if sufficient blood was available. After 72 hours of drying time, slides were dehaemoglobinised in water for 5–10 minutes, and dried prior to storage and shipment to Australia. One set of slides was stained with Giemsa according to WHO-recommended methods [10] and the second set left unstained. Each slide was examined independently for Mf by a trained parasitologist in Australia, one examining stained slides and the other the unstained slides, before comparing results. A participant was considered as Mf-positive if Mf were observed in one or both slides. Mf counts per 60uL were averaged between the two slide readers and extrapolated to density per mL.

## Clinical examination for lymphedema

During household surveys, examinations were conducted on participants aged ≥15 years to identify signs of swelling in upper or lower limbs (limited to below knee). Examinations were conducted while participants were standing or sitting, and participants did not have to lie down or undress. If swelling was present, severity was recorded using a modified international grading scale [26] that included presence of shallow/deep skin folds, knobs, mossy lesions, inter-digital lesions, entry lesions, whether the swelling goes away overnight, and impact on mobility and daily activities. Due to logistic and privacy reasons, a hydrocele survey was not conducted.

## Timing of survey

The baseline survey was planned to be completed prior to the first round of triple drug MDA, which was conducted in the last two weeks of August 2018. Due to logistic reasons beyond the research team's control, the survey was delayed and took place 7–11 weeks post-MDA. Since LF Ag persists for at least months after treatment, the results of the survey are expected to provide an accurate measure of pre-MDA Ag prevalence. However, Mf usually clear rapidly after treatment, and the Mf prevalence reported here is expected to be significantly lower than pre-MDA levels.

## Statistical analysis

Data were analysed using Stata statistical software (StataCorp, Version 15.1, College Station, TX). Summary statistics for Ag and Mf prevalence were calculated for age groups, gender, PSUs (random vs purposive), and region. Summary statistics with 95% confidence intervals were used to describe the prevalence of Ag and Mf for the two main age groups: 5–9 year-olds (from convenience and household surveys combined) and ≥10 year-olds (from household surveys), as well as an overall prevalence estimate for all ages ≥5 years. Adjustment for selection probability and clustering was based on the 2016 Samoa Census [24] and performed using the 'svyset' command in Stata with PSU as the unit of clustering. The 2016 census populations ranged from 613 to 4289 in random PSUs, and 136 to 4260 in purposive PSUs. The number of households ranged from 88 to 663 in random PSUs and 18 to 613 in purposive PSUs. Age group and gender standardized weights were applied using information from the 2016 Samoa Census [24]. Prevalence estimates for the two main age groups were adjusted for selection probability and clustering and standardized for gender but not age. The prevalence estimates for all ages ≥5 years were adjusted for selection probability and clustering, and standardized for gender (except when comparing between genders) and age using 5-year age bands. Further detail on standardization and adjustments, including the values used, are given in S1 Text, S1 and S2 Tables.

Associations between Ag prevalence in 5–9 year-olds and ≥10 year-olds were examined using linear regression. We evaluated the presence of Ag-positive 5–9 year-olds in a PSU as an indicator of Ag prevalence in those aged ≥10 years (using prevalence thresholds of 1%, 2%, 5%, and 10%) by using sensitivity, specificity, positive predictive value (PPV) and negative predictive value (NPV). In other words, if we only tested children aged 5–9 years and used the results as indicators of villages with high prevalence in older ages (defined using the above thresholds), how well would we have performed? Sensitivity was defined as the % of PSUs correctly identified as having overall Ag prevalence above each of the defined thresholds; specificity as the % of PSUs correctly identified as having Ag prevalence below the defined thresholds; PPV as the % of PSUs with Ag-positive children where Ag prevalence truly exceeded the thresholds; and NPV as PSUs without Ag-positive children where Ag prevalence was truly below the thresholds.

Clustering of Ag-positive and Mf-positive individuals were examined using multilevel hierarchical modelling to generate intraclass correlation coefficients (ICCs) with corresponding 95% confidence intervals, using the three levels of region (n = 4), PSU (n = 35), and household (n = 495) as random effects (Stata command *melogit*). Age and gender were included in the models as fixed effects. Children from the convenience survey were not included in ICC calculations because household-level data were not available.

## Spatial data and mapping

Spatial data on country, island, region and village boundaries in Samoa were obtained from the Pacific Data Hub (pacificdata.org) and DIVA-GIS (diva-gis.org). Geographic information systems software ArcMap (v10.6, Environmental Systems Research Institute, Redlands, CA) was used to manage spatial data and produce maps.

## Results

### Demographics of study population

A total of 3940 participants aged ≥5 years were recruited for the LF seroprevalence study. All participants aged ≥10 years (n = 1998) were recruited through household surveys. Participants aged 5–9 years (n = 1942) were recruited through a combination of household (n = 400) and convenience surveys (n = 1542). There were 3413 participants from randomly selected and 527 from purposively selected PSUs (Table 1).

Overall there were 48.9% male and 51.1% female participants; a larger proportion of males were surveyed in those aged 5–9 years, while there was a larger proportion of females in those aged 20 to 59 years. The study population included a disproportionately large number of children aged 5–9 years due to the survey design (Fig 2). In randomly selected PSUs, 49.3% of participants were male, but there was a significantly higher percentage of males in participants aged 5–9 years compared to those aged ≥10 years (52.9% vs 45.8%, $Chi^2$ = 16.94, $p<0.001$). Similarly, in the purposively selected PSUs, 46.5% of participants were male, and the percentage of males was higher in those aged 5–9 years than in those aged ≥10 years (53.1% vs 40.2%, $Chi^2$ = 8.8, $p<0.01$). In randomly selected PSUs, the proportion of participants from each region was reflective of the population distribution of Samoa.

A total of 499 households were surveyed (437 in random PSUs and 62 in purposive PSUs), and valid Ag results were available from 495 (99.2%) households. The mean numbers of households per PSU was similar between random (14.9) and purposive (13.3) PSUs, and close to the target sample sizes of 15 households per PSU. The mean number of participants per PSU (including convenience survey children) was similar between random (113.8) and purposive (106.4) PSUs, and close to the target sample size of 114 persons per PSU (Table 2).

**Table 1. Demographics of study population in randomly versus purposively selected PSUs.**

| | Randomly selected PSUs n (%) | Purposively selected PSUs n (%) |
|---|---|---|
| **Age group** (years) | 3413 | 527 |
| 5–9 | 1686 (49.4) | 256 (48.6) |
| ≥10 | 1727 (50.6) | 271 (51.4) |
| **Sex** | | |
| Male | 1682 (49.3) | 245 (46.5) |
| Female | 1731 (50.7) | 282 (53.5) |
| **Region** | | |
| AUA | 668 (19.6) | - (-) |
| NWU | 1286 (37.7) | 342 (64.9) |
| ROU | 810 (23.7) | 75 (14.2) |
| SAV | 649 (19.0) | 110 (20.9) |

AUA = Apia Urban Area, NWU = Northwest Upolu, ROU = Rest of Upolu, SAV = Savai'i.

## Antigen prevalence by age and gender

Of the 3940 participants in all PSUs (3413 in random and 527 in purposive), FTS was performed on blood samples from 3883 (98.6%). For the 57 (1.4%) where FTS was not done, reasons included 'no consent' (n = 17), 'not enough blood' (n = 16), and 'no reason given' (n = 24). Among those who had FTS performed, 31 (0.8%) had invalid results due to no (or slow) flow of blood or lack of control line, resulting in 3852 participants (99.2% of those tested, and 97.8% of all participants) with valid FTS results.

In the 30 randomly selected PSUs, valid FTS results were available for 3333 (97.7%) participants (1668 aged 5–9 years and 1665 aged ≥10 years). Of these, positive Ag was identified in 24 (1.4%) of those aged 5–9 years and 67 (4.0%) of those aged ≥10 years. Adjusted Ag

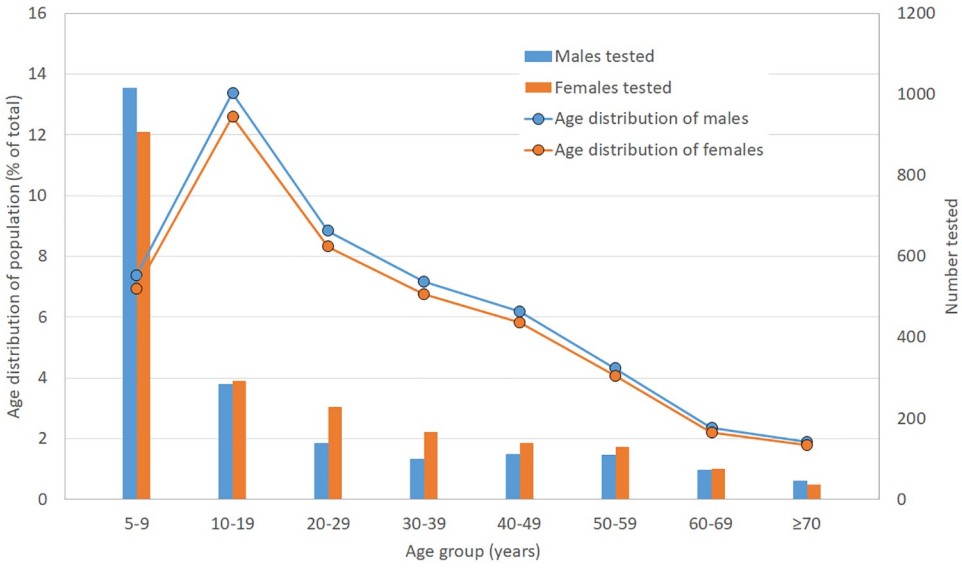

**Fig 2. Age distribution of participants by gender (randomly selected PSUs only), compared to age structure of the general population in Samoa.**

**Table 2. Characteristics of study populations in randomly versus purposively selected PSUs, by number of households, household size, age groups, and survey location (convenience or household surveys).**

|  | Randomly Selected PSUs | Purposively Selected PSUs |
|---|---|---|
| Total PSUs | 30 | 5 |
| **Total households sampled** | 437 | 62 |
|  | Mean (range) | Mean (range) |
| **Households per PSU** | 14.9 (9–20) | 13.3 (6–15) |
| **Household size (persons aged ≥5 years)** | 4.8 (1–26) | 5.2 (1–13) |
| **Participants per PSU (all ages):** | 113.8 (99–128) | 106.4 (75–117) |
| **Age 5–9 years (total)** | 56.2 (40–75) | 51.2 (36–59) |
| Convenience survey | 44.5 (29–61) | 41.2 (30–49) |
| Household survey | 11.7 (5–22) | 10.0 (6–13) |
| **Age ≥10 years** |  |  |
| Household survey | 57.6 (43–73) | 54.2 (39–59) |

prevalence was lower in participants aged 5–9 years (1.2%, 95% CI 0.8–2.1%) than those ≥10 years (4.7%, 95% CI 3.1–7.0%) ($p<0.0001$). Adjusted overall Ag prevalence in all ages ≥5 years in random PSUs was 4.0% (95% CI 2.8–5.6%); prevalence was higher (although not statistically significantly so) in males (4.7%) than females (3.1%) ($p = 0.06$); differences between genders were more pronounced in those aged >20 years (Fig 3).

In the five purposively selected PSUs, valid FTS results were available for 519 (98.5%) participants. Of these, positive Ag were identified in 4 (1.6%) participants aged 5–9 years and in 27 (10.2%) aged ≥10 years. Adjusted Ag prevalence was lower in those aged 5–9 years (2.1%, 95% CI 1.0–4.3%) compared to those aged ≥10 years (11.4%, 95% CI 7.9–16.1%) ($p<0.05$). Adjusted Ag prevalence in all ages ≥5 years in purposive PSUs was 10.0% (95% CI 7.4–13.4%),

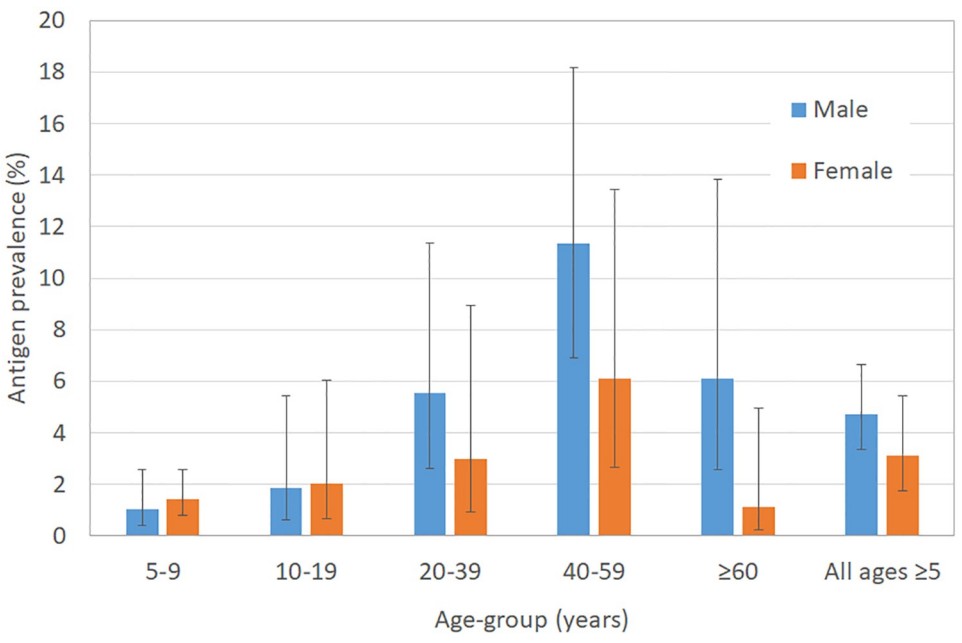

**Fig 3. Antigen prevalence in randomly selected PSUs by age and gender (adjusted for sampling design).**

significantly higher than in randomly selected PSUs (4.0%, 95% CI 2.8–5.6%) ($p<0.01$). The difference in Ag prevalence between the two age groups was much greater in purposive than random PSUs (Fig 4). Adjusted Ag prevalence for participants aged 5–9 years, ≥10 years, and all ages ≥5 years in randomly selected PSUs (combined and by region) versus purposively selected PSUs are summarised in Fig 4.

### Antigen prevalence by region and PSU

Adjusted Ag prevalence in randomly selected PSUs varied between and within regions, and was higher in those aged ≥10 years compared to 5–9 years (Figs 4 and 5 and S1 Fig). In randomly selected PSUs, adjusted Ag prevalence in those aged 5–9 years was highest in ROU (1.8%), followed by NWU (1.2%), SAV (1.1%), and AUA (1.0%), but differences were not statistically significant (Fig 4). Ag prevalence in those aged ≥10 years was highest in NWU (5.7%), followed by AUA (5.1%), SAV (3.6%) and ROU (2.0%) but differences were not statistically significant (Fig 4).

Ag-positive participants were identified in 23 of the 30 randomly selected PSUs, with a range of 1–13 Ag-positive persons per PSU. Adjusted Ag prevalence in all ages ≥5 years in randomly selected PSUs ranged from 0% (1-sided 97.5% CI 0–3.69%) to 10.3% (95% CI 5.9–17.6%). Ag-positive people were identified in all five purposive PSUs, ranging from two to 12 per PSU; adjusted PSU-level prevalence in all ages ≥5 years ranged from 3.8% (95% CI 1.3–10.8%) to 20.0% (95% CI 15.3–25.8%) (Fig 5 and S1 Fig). Maps of adjusted Ag prevalence in those aged 5–9 years and ≥10 years are provided in S2 and S3 Figs.

### Association between Ag prevalence in participants aged 5–9 years and ≥10 years

Ag-positive participants aged 5–9 years were identified in 19 (54.3%) of the 35 PSUs, and four (11.4%) PSUs had more than one Ag-positive child aged 5–9 years (two Ag-positive children

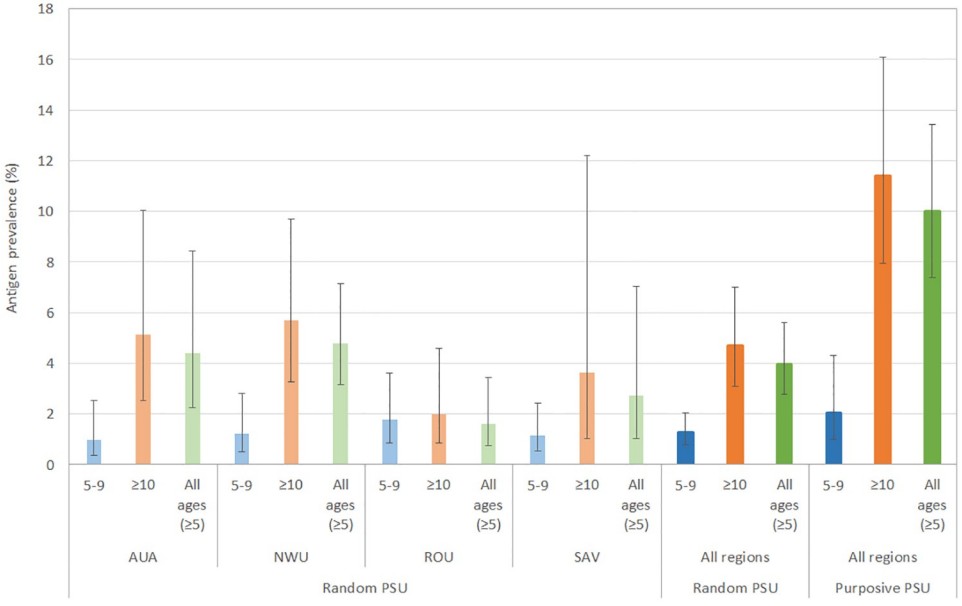

**Fig 4. Adjusted antigen prevalence in randomly selected (by region and total) and purposively selected PSUs by age groups (years).**

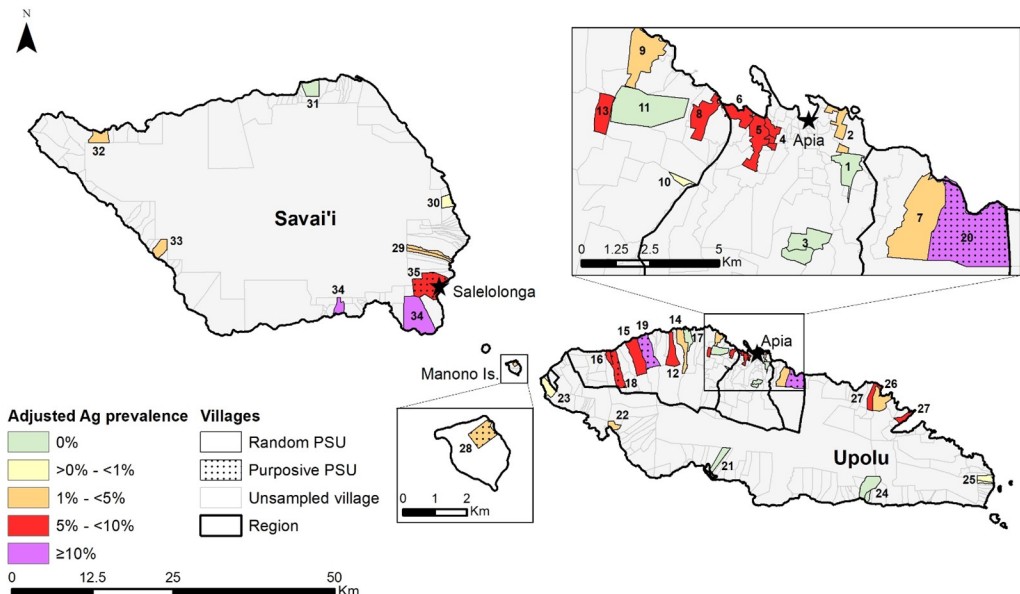

**Fig 5. Adjusted antigen prevalence for all ages ≥5 years in randomly and purposively selected PSUs.**

in three PSUs, and seven Ag-positive children in one PSU). At the PSU level, linear regression showed significant association but poor correlation between Ag prevalence in 5–9 year-olds and those aged ≥10 years ($R^2$ 0.1459, $p<0.05$) (Fig 6).

Table 3 shows the sensitivity, specificity, PPV, and NPV for using the presence of Ag-positive 5–9 year-olds as indicators of villages with adjusted Ag prevalence of greater than 1%, 2%, 5%, and 10% in those aged ≥10 years. For an Ag prevalence threshold of >1% (reflective of programmatic targets), the presence of at least one Ag-positive 5–9 year-old had moderate sensitivity (68.6%) and NPV (50.0%) for identifying all PSUs in this category, but the specificity (72.7%) and PPV (84.2%) were relatively high. In other words, testing 5–9 year-olds is unlikely to help identify all areas with >1% Ag prevalence or provide reassurance that Ag prevalence is <1%, but in areas where Ag-positive children are found, there is high probability that there is ongoing transmission. The presence of at least one Ag-positive child aged 5–9 years was more sensitive for identifying PSUs with high Ag prevalence in older ages (e.g. 83.3% for PSUs with Ag prevalence of >10%), but by this stage, resurgence would likely be already well established. The presence of more than one Ag-positive child aged 5–9 years was highly specific (>93%) for identifying areas of transmission at all prevalence thresholds used, and had 100% PPV that Ag prevalence in older age groups was >2%.

## Microfilaria prevalence

Of 122 Ag-positive individuals, slides were available for 121; 90 of 91 in randomly selected PSUs and all 31 in purposively selected PSUs. Only one slide (stained) was available for 9 of the 122 Ag-positive persons, and in one case no slides were available because the blood sample was misplaced. For the other 112 participants, there was good correlation in reported Mf counts (r = 0.859) between the two slide readers, and reports of Mf-positivity was concordant with one exception where one reader identified 9 Mf on the slide (60 μL) while the other reader did not identify any.

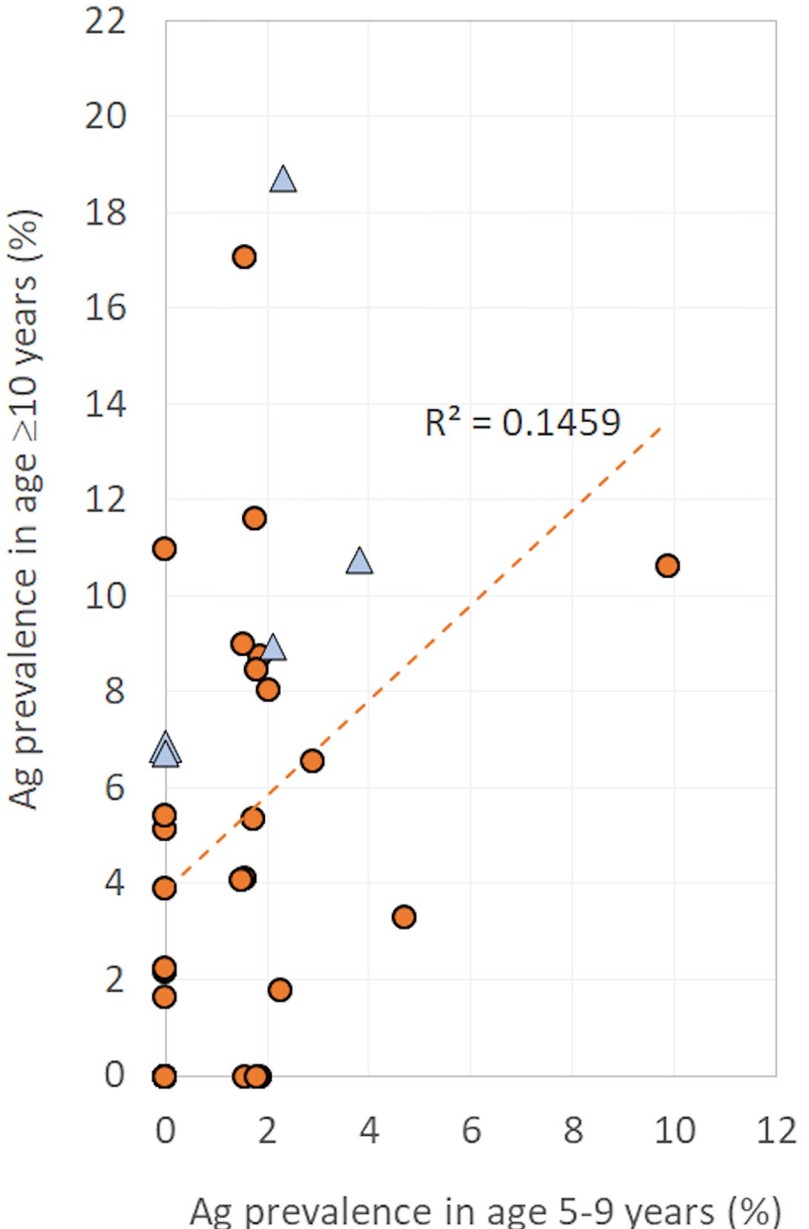

**Fig 6. Correlation between adjusted Ag prevalence in participants aged 5–9 years and ≥10 years.** Orange circles and blue triangles represent randomly selected and purposively selected villages, respectively.

For random PSUs, 13 (14.4%) slides were Mf-positive; overall adjusted Mf prevalence was 0.6% (95% CI 0.3–1.0%). In purposive PSUs, 5 (16.1%) slides were Mf-positive and adjusted Mf prevalence was 1.7% (95% CI 0.7–4.1%) (Table 4). The distribution of Mf counts ranged from 16.7/mL to 1041.7/mL, with geometric mean of 128.2/mL in the random PSUs and 148.4/mL in the purposive PSUs. Mf prevalence was significantly higher in participants aged ≥10 years than in those aged 5–9 years in both randomly (Chi$^2$ = 9.37, $p<0.01$) and purposive PSUs (Chi$^2$ = 4.88, $p<0.05$).

**Table 3. Sensitivity, specificity, positive predictive value (PPV) and negative predictive value (NPV) of i) the presence of at least one Ag-positive child aged 5–9 years and ii) more than one Ag-positive child aged 5–9 years, as indicators of villages with adjusted Ag prevalence in ≥10 year-olds of greater than 1%, 2%, 5%, and 10%.**

| Indicator | Adjusted Ag prevalence in age ≥10 years in PSU | Number of PSUs (%) | Sensitivity (%) | Specificity (%) | PPV (%) | NPV (%) |
|---|---|---|---|---|---|---|
| i) PSU with at least one Ag-positive child aged 5–9 years | >1% | 25 (71.4) | 68.6 | 72.7 | 84.2 | 50.0 |
| | >2% | 23 (65.7) | 70.0 | 66.7 | 73.7 | 62.5 |
| | >5% | 17 (48.6) | 70.6 | 61.1 | 63.2 | 68.8 |
| | >10% | 3 (8.6) | 83.3 | 51.7 | 26.3 | 93.8 |
| ii) PSU with more than one Ag-positive child aged 5–9 years | >1% | 24 (68.6) | 16.0 | 100 | 100 | 32.3 |
| | >2% | 20 (57.1) | 17.4 | 100 | 100 | 38.7 |
| | >5% | 14 (40.0) | 17.6 | 94.4 | 75.0 | 54.8 |
| | >10% | 3 (8.6) | 33.3 | 93.1 | 50.0 | 87.1 |

Grey: 0–25.0%, light blue: 25.1–50.0%, medium blue: 50.1–75.0%, dark blue: 75.1–100%.

## MDA participation in Ag and Mf positive persons

In the 2018 round of triple-drug MDA, there was no difference in the proportion who reported taking MDA in Ag-positive (n = 122, 87.7%) versus Ag-negative participants (N = 3725, 90.9%,) (Chi$^2$ = 1.42, $p$ = 0.23). Of the 18 Mf-positive persons, 14 (77.8%) reported taking MDA compared to 3828 (90.9%) of those who were Mf-negative, but this difference was not statistically significant (Chi$^2$ = 3.67, $p$ = 0.06).

## Clustering of Ag and Mf-positive participants

Of the 499 households included in the analyses, 495 had at least one person with valid FTS results. The remaining four households only had one resident aged ≥5 years, and these participants either did not have FTS done or the result was invalid. Of the 495 households, Ag-positive participants were identified in 77 (15.6%) households; 59 (11.9%) included one Ag-positive member, 12 (2.4%) included two, five (1.0%) included three, and one (0.2%) included five Ag-positive members. For Ag-positive participants, clustering as measured by ICC was highest at the household level (0.46), followed by PSU (0.18) and region (0.01). Mf-positive

**Table 4. Antigen and Mf positives in randomly versus purposively selected PSUs by age group.**

| | Randomly selected PSUs | | | Purposively selected PSUs | | |
|---|---|---|---|---|---|---|
| Age group* | 5–9 years | ≥10 years | All ages ≥5 years | 5–9 years | ≥10 years | All ages ≥5 years |
| Valid FTS results | 1668 | 1665 | 3333 | 255 | 264 | 519 |
| Ag-positive, n (%) | 24 (1.4%) | 67 (4.0%) | 91 | 4 (1.6%) | 27 (10.2%) | 31 |
| Adjusted Ag prevalence (%, 95% CI) | 1.3% (0.8–2.1%) | 4.7% (3.1–7.0%) | 4.0% (2.8–5.6%) | 2.1% (1.0–4.3%) | 11.4% (7.9–16.1%) | 10.0% (7.4–13.4%) |
| Mf slides# available | 24 | 66 | 90 | 4 | 27 | 31 |
| Mf positive, n (% of slides) | 1 (4.2%) | 12 (18.2%) | 13 | 0 (0%) | 5 (18.5%) | 5 |
| Mf prevalence % of valid FTS (N or 95% CI)* | 0.1% (1668) | 0.7% (1664)^ | 0.6% (0.3–1.0%) | 0.0% (255) | 1.9% (264) | 1.7% (0.7–4.1%) |

* All results adjusted for sampling design; results for age 5–9 years and ≥10 years standardized for gender; results for all ages ≥5 years standardized for both age and gender.

# Mf slides were only prepared for positive FTS samples.

^ Denominator for overall prevalence is the sum of those who were FTS negative (assumed Mf negative, no slide done) and those who were FTS positive with slide available. Thus, the one FTS positive person with no slide available was excluded from analyses.

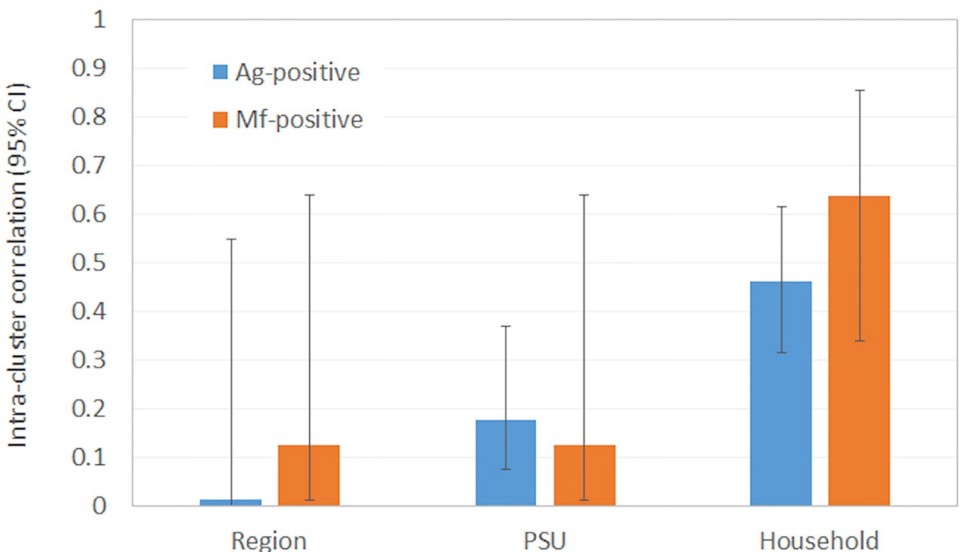

**Fig 7. Intra-cluster correlation for Ag-positive and Mf-positive individuals at region, PSU, and household levels.**

participants were identified in 14 (2.8%) households; 12 (2.4%) households included one Mf-positive member, one (0.2%) had two Mf-positive residents, and one (0.2%) had three Mf-positives. For Mf-positive participants, ICC was also highest at the household level (0.63) but were the same at PSU and region (0.12) levels. For both Ag and Mf, clustering was more intense at the household level, but differences were not statistically significant (Fig 7).

## Lymphedema

Of the 1622 participants aged ≥15 years who were examined for lymphedema, 26 (1.6% had swelling of at least one limb; of these, 20 (1.2%) had unilateral swelling of one leg, and two (0.1%) had unilateral swelling of one arm. Detailed notes and photographs taken by the field team were reviewed by a clinician, and none of the 26 (0%, 1-sided CI 97.5% CI 0–0.2%) were thought to have LF-related lymphedema. Clinical management of LF was outside of the scope of this study, and individuals with chronic complications of LF were advised to seek care through local medical services.

## Discussion

Our study confirmed resurgence of LF in Samoa seven years after the last round of nationwide MDA and one year after targeted MDA in NWU. Through a community-based survey of randomly selected PSUs conducted 7–11 weeks after the first round of triple-drug MDA, we identified 24 Ag-positive children aged 5–9 years (adjusted Ag prevalence 1.3%), including a five year-old who was Mf-positive. A further four Ag-positive 5–9 year-old children were identified in the purposively selected PSUs (adjusted Ag prevalence 2.1%), but none of them were Mf-positive. In household surveys of community members aged ≥10 years, 67 and 27 Ag-positive persons were identified in random and purposive PSUs, with adjusted Ag prevalence of 4.7% and 11.4% respectively. Ag-positive participants were identified in 23 of the 30 random PSUs, and all five of the purposive PSUs. The presence of Ag-positive persons, including children aged 5–9 years, in multiple PSUs across all regions of Samoa indicate widespread transmission in recent years.

In both random and purposive PSUs, Ag prevalence was higher in older age groups compared to children aged 5–9 years; this finding is consistent with reports from multiple countries that Ag prevalence is generally higher in older persons [27,28]. As previously observed in neighbouring American Samoa [14], highest Ag prevalence was observed in adult males, but did not differ between genders in young children. Similar findings were observed in our study, where Ag prevalence in males aged ≥20 years was more than twice as high as females, but there was little difference between genders in younger ages (Fig 3). Differences between genders in the smaller age categories in Fig 3 were not statistically significant, but it should be noted that the survey was not powered to detect differences between these subgroups. Higher prevalence in males is widely observed in studies around the world, but the reasons for this finding are unclear. Possible explanations include lower MDA participation rates [29,30], more time spent outdoors for work and recreation, or hormonal factors [31].

We expect the timing of the survey to have affected Mf prevalence and to a lesser degree Ag prevalence. Mf clearance can occur very rapidly after treatment; Thomsen *et al.* reported Mf clearance in 91.7% of participants (n = 12) one week after IDA [32], and McCarthy *et al.* reported that 95.8% of participants (n = 24) were Mf-negative one week after taking DEC alone [33]. Previous studies have reported varying rates of Ag clearance after treatment with IDA. A study in Côte d'Ivoire found that after a single dose of IDA, mean circulating filarial Ag (measured by ELISA) reduced by 70% at six months and 75% at 12 months, while FTS score reduced by 50% at 6 months and 55% at 12 months [34]. Similar results were observed in PNG, where a 50% reduction in Og4C3 Ag level was observed at 12 months [32]. Studies in PNG [35,36] and Haiti [37] reported that after a single dose of IDA, 95%-97% and 79.5% of Ag-positive participants were still positive after 12 months, respectively. These findings suggest that after treatment with IDA, Ag-positivity is likely to persist for months in the majority of people. None of the studies that used a single dose of two-drug or three-drug combinations investigated Ag clearance rates around the 7–11 week period post-treatment (when our survey in Samoa was conducted), so there is currently insufficient knowledge about Ag clearance in the immediate post-MDA period for us to accurately estimate true prevalence at baseline. However, we can be confident that the baseline Ag and Mf prevalence before the first round of triple-drug MDA in Samoa would have been at least as high as our reported values.

Given that the 2018 baseline survey was conducted 7–11 weeks after the first round of triple-drug MDA, it is surprising that our study identified 18 Mf-positive persons, 14 (78%) of whom reported taking the MDA. As mentioned earlier, Mf prevalence was likely to have been higher before MDA, but it was not possible for our study to provide an accurate estimate. Ag persists for at least months after MDA [33], but clearance of Mf is expected to occur within one week of triple-drug treatment [32]. Possible explanations for persistence of Mf despite high reported MDA adherence include inadequate dosing for body weight, inadequate drug concentration despite adequate dose ingestion, inaccurate reporting by participants (including social desirability bias, i.e. reporting what they perceive that interviewers want to hear), or inaccurate reporting by survey staff. Drug resistance is a possible explanation, but our preliminary investigations suggested that this was unlikely in Samoa. Follow–up of these participants through more in-depth interviews, directly-observed weight-appropriate dosing, and repeat Ag and Mf testing is under way.

Our study identified significant variation in Ag prevalence between PSUs and regions, even within the relatively small and isolated islands of Samoa. We have previously observed similar findings in the even smaller islands of American Samoa [12–14]. Post-MDA surveys in other small populations such as Sri Lanka, Vanuatu, and Tonga have also reported similar spatial heterogeneity [38–40]. Spatial heterogeneity in Ag prevalence presents challenges in surveillance and monitoring because random sampling may hit/miss hotspots and overestimate/

underestimate true prevalence, and aggregation of results in large EUs may provide an overall low prevalence even if hotspots exist. Failure to accurately determine true prevalence and/or identify hotspots could contribute to the risk of resurgence. Ag prevalence in purposive PSUs was higher than in random PSUs, indicating the importance of local knowledge about areas where ongoing transmission is likely, and where more intensive targeted surveillance may be warranted. Our study identified more intense clustering of Ag-positive and Mf-positive persons at the household level than at PSU or region levels. This finding is consistent with our previous surveys in American Samoa [41] and supports targeted surveillance of household members of positive persons, regardless of how they were identified. Targeted testing of near neighbours may also be considered.

At the PSU level, Ag-prevalence in children aged 5–9 years was poorly correlated with Ag prevalence in older ages (≥10 years) and had poor sensitivity for identifying locations with >1% Ag prevalence in all ages, despite the 5–9 year-olds making up approximately 50% of the sampled populations. Sampling older age groups would therefore provide more accurate estimates of overall prevalence, and be more useful for identifying residual hotspots in a cost-effective manner. This finding supports our conclusions from previous studies in American Samoa, which found that when compared to a school-based TAS of children aged 6–7 years, community surveys of older persons provided a better indication of overall Ag prevalence [14]. However, in this study, the presence of more than one Ag-positive child in a community provided 100% specificity and PPV that Ag prevalence in those aged ≥10 years was >2%, and >93% specificity that Ag prevalence was >10%, indicating that more intense interventions would be strongly recommended in these communities.

The results of this paper provide further support for the concept of a multi-stage surveillance strategy suggested by Lau *et al.* [41], starting with a population representative survey followed by more intensive targeted sampling of high-risk groups (e.g. household members, adult males) and high prevalence locations. Heterogeneity (clustering) observed between and within PSUs and households also leads to consideration of different sampling strategies including spatially explicit sampling and prediction [42], adaptive or snowball sampling [43], the use of markers that are potentially more sensitive than Ag or Mf (e.g. anti-filarial antibodies [27,41,44]), molecular xenomonitoring of mosquitoes [45–48], or combinations of these.

The reasons for LF persistence and resurgence in Samoa are unclear. As detailed in the introduction, previous surveys indicated that multiple rounds of MDA did not reduce prevalence thresholds to below recommended levels. Although there have been many surveys in Samoa over the years, none have specifically addressed reasons for ongoing transmission despite completing the recommended rounds of MDA. Potential explanations include local factors such as tropical climate and outdoor lifestyle, intense transmission (highly efficient mosquito vectors, both day and night biting), and travel and migration (leading to missed MDA, importation of parasites by travelers, and subsequent spread within the country). Prior to the triple-drug MDA in 2018, the last nationwide MDA was distributed in 2011 [21]. Although there were two targeted MDAs in NWU in 2015 and 2017, high connectivity between different regions of Samoa (especially with Apia) means that parasites could have spread between regions after 2011. Programmatic factors that might have contributed to resurgence include low coverage, systematic non-compliance, or target thresholds not low enough for the local setting. To optimize future success, programs will require surveillance strategies that provide more accurate estimates of prevalence (e.g. testing older age groups, molecular xenomonitoring) and are more sensitive for identifying residual hotspots (e.g. spatial sampling strategies).

Samoa was proactive and expedient in addressing the resurgence of LF, with the first round of IDA distributed in 2018 [23], and the second round planned for 2020/2021 (delayed by

measles outbreak and COVID-19). Follow-up surveys will assess the impact of two rounds of triple-drug MDA on transmission. Other surveillance strategies being investigated in Samoa include spatial sampling strategies [42] and molecular xenomonitoring.

## Supporting information

**S1 Text. Adjustment for sampling design.**
(DOCX)

**S1 Table. Parameters used for adjustments and standardization.**
(DOCX)

**S2 Table. Adjustments and standardization used for different estimates.**
(DOCX)

**S1 Fig. Adjusted antigen prevalence for all ages ≥5 years in randomly and purposively selected PSUs.**
(TIF)

**S2 Fig. Adjusted antigen prevalence for 5–9 year-olds in randomly and purposively selected PSUs.**
(TIF)

**S3 Fig. Adjusted antigen prevalence for ≥10 year-olds in randomly and purposively selected PSUs.**
(TIF)

**S1 Checklist. STROBE checklist for cross-sectional studies.**
(DOCX)

## Acknowledgments

We would like to thank all the staff at the Samoa Ministry of Health who supported the many different aspects of the study. We especially thank Miriama Asoiva who provided valuable advice on local logistics and cultural sensitivities, and assistance with obtaining permissions to conduct village visits. We also thank Tile Ah Leong-Lui, Fuatai Maiava and Siatua Loau for sharing their knowledge about the LF elimination program in Samoa, and to Fuatai for making it possible for nurses to assist with the household surveys. We sincerely thank Tautala Maula, the general secretary of the Samoa Red Cross, and her team (especially Babey Suniula, Nixon Mataia, Brenda Koon Wai You, Alesi Mataia, and Shem Lepale) for their enthusiastic and untiring support with fieldwork, village visits, and laboratory work; this survey would not have been possible without their hard work and dedication. We greatly appreciate all the support and advice provided by Rasul Baghirov and Lepaitai Hansell at the WHO country office in Samoa, and thank them for generously sharing their wisdom. We also thank the Australian volunteers and students (Gabriela Willis, Meru Sheel, Benjamin Dickson) who assisted with fieldwork and data management, and technical advice provided by Patrick Lammie (Task Force for Global Health) and Kimberly Won (US Centers for Disease Control and Prevention). We thank the NIH/NIAID Filariasis Research Reagent Resource Center (www.filariasiscenter. org) for supplying positive controls for the Filariasis Test Strips.

## Author Contributions

**Conceptualization:** Colleen L. Lau, Sarah Sheridan, Katherine Gass, Patricia M. Graves.

**Data curation:** Colleen L. Lau, Kelley Meder, Helen J. Mayfield, Therese Kearns, Brady McPherson, Shannon M. Hedtke, Sarah Sheridan, Patricia M. Graves.

**Formal analysis:** Colleen L. Lau, Kelley Meder, Helen J. Mayfield, Patricia M. Graves.

**Funding acquisition:** Colleen L. Lau, Katherine Gass, Patricia M. Graves.

**Investigation:** Colleen L. Lau, Kelley Meder, Helen J. Mayfield, Therese Kearns, Brady McPherson, Shannon M. Hedtke, Sarah Sheridan, Patricia M. Graves.

**Methodology:** Colleen L. Lau, Sarah Sheridan, Katherine Gass, Patricia M. Graves.

**Project administration:** Colleen L. Lau, Take Naseri, Robert Thomsen, Katherine Gass, Patricia M. Graves.

**Resources:** Colleen L. Lau, Katherine Gass, Patricia M. Graves.

**Supervision:** Colleen L. Lau, Therese Kearns, Take Naseri, Robert Thomsen, Sarah Sheridan, Patricia M. Graves.

**Visualization:** Colleen L. Lau, Helen J. Mayfield, Patricia M. Graves.

**Writing – original draft:** Colleen L. Lau, Kelley Meder, Patricia M. Graves.

**Writing – review & editing:** Colleen L. Lau, Kelley Meder, Helen J. Mayfield, Therese Kearns, Brady McPherson, Take Naseri, Robert Thomsen, Shannon M. Hedtke, Sarah Sheridan, Katherine Gass, Patricia M. Graves.

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
