## [Decision Letter · Decision Letter 0]

1 Oct 2020

Dear Dr Lau,

Thank you very much for submitting your manuscript "Lymphatic Filariasis Epidemiology in Samoa in 2018:   Geographic Clustering and Higher Antigen Prevalence in Older Age Groups" for consideration at PLOS Neglected Tropical Diseases. As with all papers reviewed by the journal, your manuscript was reviewed by members of the editorial board and by several independent reviewers. The reviewers appreciated the attention to an important topic. Based on the reviews, we are likely to accept this manuscript for publication, providing that you modify the manuscript according to the review recommendations. 

Please address the reviewers comments.

Sincerely,

Keke Fairfax, PhD

Deputy Editor

Keke Fairfax

Deputy Editor

Please address the reviewers comments.

Reviewer's Responses to Questions

**Key Review Criteria Required for Acceptance?**

**Methods**

-Are the objectives of the study clearly articulated with a clear testable hypothesis stated?

-Is the study design appropriate to address the stated objectives?

-Is the population clearly described and appropriate for the hypothesis being tested?

-Is the sample size sufficient to ensure adequate power to address the hypothesis being tested?

-Were correct statistical analysis used to support conclusions?

-Are there concerns about ethical or regulatory requirements being met?

Reviewer #1: The sampling approaches were mainly appropriate and well executed. 

Line 105: Ivermectin is used for treating onchocerciasis. The triple regimen is indicated for areas where onchocerciasis is not endemic. Can the author add a brief comment why ivermectin was included in the triple regimen and then indicated for areas not endemic for onchocerciasis? 

Line 112: “achievement of GPELF”

Line 195: a reference or a rationale is needed for the choice of the sample size determination. Ideally, >80% power is used to determine sample size.

Instead of convenience sampling for logistic reason, it would have been ideal to reduce the n to sample more household in random fashion. However, the other sampling approaches were good.

Hydrocele index is a good indicator. This was not sampled for logistic reasons. However, in private interviews, a simple question posed with option not to respond will be useful in the light of determining the burden of disease.

Line 283: If the TAS was performed post MDA, then the Antibody study will be an important component of this publication. Also as indicated by the authors, a post MDA study on Mf is not useful as a baseline data.

Reviewer #2: see attached

Reviewer #3: The objectives were well defined and easy to understand. The study methodology is well detailed, in a structured way and meets the proposed objectives. The criteria for choosing the participants, the selection of the households and children's survey also seems correct.

It presents a robust statistical analysis, according to the type of study.

Ethical issues were granted

**Results**

-Does the analysis presented match the analysis plan?

-Are the results clearly and completely presented?

-Are the figures (Tables, Images) of sufficient quality for clarity?

Reviewer #1: The baseline study readily identified hotspots, but only based on (by purposive sampling) previous knowledge of ongoing transmission in these locations. The rationale for purposive sampling is unclear as this may have skewed the overall reported prevalence. However, a separate analysis was done for randomly selected units, a more useful indices that the purposive sampling and unfortunately the overall prevalence. While this is an interesting finding, reidentifying a hotspot in an already known high transmission location is only academic. The data from random PSUs arrived at the same conclusion that Ag prevalence was higher in the >10yo. This too is an expected outcome since the higher age groups are more likely to be involved in outdoor activities that lead to exposure to infected mosquitoes. And for a chronic disease like LF, time is always a factor. 

There is need to add a comment on the findings of other evaluation studies that may have identified why 10 rounds of MDA failed in Samoa while fewer rounds eliminated LF in some locations. While the authors added their own speculations on why MDA failed over time in Samoa, these were not from scientific studies. If this information is a product of interviews and some form of organized data collection approach, it needs to be stated. Also, previous TAS found that while only NWU failed the first TAS in 2013, all the EUs failed TAS in 2017. This is striking and needs to be studied or commented on to put the current MDA/study in perspective. Such study is important both in planning and evaluating the current MDA.

The authors showed confidence intervals without p-values in all the important data. Although this study is more descriptive, the p-values will orient the reader to the significance of the identified differences and associations. 

Line 337: Please check the sentence “…... while a there..”

Line 553: change ‘it’s’ to ‘it is’ or ‘it was’

For all figures, please provide a caption. In addition, move the figure labels below the figure.

Reviewer #2: see attached

Reviewer #3: The results are clear and well supported by the tables and figures. The most relevant data were clearly presented . The analysis presented corresponds to the defined plan.

**Conclusions**

-Are the conclusions supported by the data presented?

-Are the limitations of analysis clearly described?

-Do the authors discuss how these data can be helpful to advance our understanding of the topic under study?

-Is public health relevance addressed?

Reviewer #1: Authors found that sampling 5-9 age group was not as useful as sampling >10 ager group in determining prevalence and cut off for stopping MDA. This appear to not be concomitant with the 6-7 ager group recommended by GPELF. More comment will be interesting on how the 6-7 age group has performed in other areas/studies, maybe a few comments on the previous study cited by the authors.

Reviewer #2: see attached

Reviewer #3: The conclusions reflect the results obtained, which at the same time were well discussed, according to the information described in the literature.

They presented the limitations of the study, which helps to understand some of the results.

This is a topic of high importance from public health point of view, as mentioned throughout the article. This work clearly contributes to the increase in knowledge of the epidemiological situation of lymphatic filariasis in Samoa, thus allowing new actions by governmental entities to control the disease.

**Editorial and Data Presentation Modifications?**

Reviewer #1: For all figures, please provide a caption. In addition, move the figure labels below the figure.

Reviewer #2: see attached

Reviewer #3: There are some aspects that should be clarified, upgrading the work. In the introduction, it is mentioned that lymphatic filariasis is “caused by three species of filarial worms (Wuchereria bancrofti, Brugia malayi and B. timori). However, it was not mentioned whether the 3 species existed in Samoa. In the results, is refered that the number of microfilaria was counted, but it is not mentioned the species of microfilaria, which from an epidemiological point of view, would be important. Is that data available?

I think that the last paragraph of the results, related to lymphedema, is not relevant for this paper, as well as the figure 8. Despite of the clinical relevance, this is not included in the estudy. 

In the references: Numbers 10, 20, 23, 28, 30, the “accession date” is missing; in numbers 22 and 24, the “accession date” and the web link are missing.

**Summary and General Comments**

Reviewer #1: Lau et al here reports “Lymphatic Filariasis Epidemiology in Samoa in 2018: Geographic Clustering and Higher Antigen Prevalence in Older Age Groups”, a baseline study prior to an MDA program with three ivermectin, DEC and albendazole. This is an important study that serves as both evaluation of previous intervention efforts which the authors allude started in the 1960s, it also serves as a reference point for the evaluation of the ongoing MDA. The report is also well written and well organized. The sampling approaches were mainly appropriate and well executed. However, a few observations on how this study can be improved have been included. While some of these comments are probably not within the scope of what can be modified at this point, efforts should be made to at least address these as limitations of the study within the paper.

Reviewer #2: see attached

Reviewer #3: I think this is a work of the greatest relevance from an academic and scientific point of view. The theme is actual, and about a debilitating disease, with a high impact on society. The article is well written, well structured and easy to understand. It presents a good introduction, well-defined objectives, appropriate methodology to the proposed objectives. The results are relevant, supported by a robust statistical analysis. A good discussion, with well-founded conclusions, supported by a good and actual Bibliographic references list

PLOS authors have the option to publish the peer review history of their article (what does this mean?). If published, this will include your full peer review and any attached files.

Reviewer #1: Yes: Evaristus Mbanefo

Reviewer #2: No

Reviewer #3: No
---

## [Editor Report · Decision Letter 1]

28 Oct 2020

Dear Dr Lau,

We are pleased to inform you that your manuscript 'Lymphatic Filariasis Epidemiology in Samoa in 2018:   Geographic Clustering and Higher Antigen Prevalence in Older Age Groups' has been provisionally accepted for publication in PLOS Neglected Tropical Diseases.

Best regards,

Keke Fairfax, PhD

Deputy Editor

Keke Fairfax

Deputy Editor

---

## [Editor Report · Acceptance letter]

11 Dec 2020

Dear Dr Lau,

We are delighted to inform you that your manuscript, "Lymphatic Filariasis Epidemiology in Samoa in 2018:   Geographic Clustering and Higher Antigen Prevalence in Older Age Groups," has been formally accepted for publication in PLOS Neglected Tropical Diseases.

Best regards,

Shaden Kamhawi

co-Editor-in-Chief

Paul Brindley

co-Editor-in-Chief
